# The prioritisation of curable sexually transmitted infections among pregnant women in Zambia and Papua New Guinea: Qualitative insights

Lisa M. Vallely[1,2], Kelvin Kapungu[3], Alice Mengi[1], Mike Chaponda[3,4], R. Matthew Chico[4], Michaela A. Riddell[1,2], Andrew J. Vallely[1,2], William Pomat[1], Eva Cignacco[5], Nicola Low[6], Angela Kelly-Hanku[1,2]*

1 Papua New Guinea Institute of Medical Research, Goroka, Eastern Highlands Province, Papua New Guinea, 2 The Kirby Institute, UNSW Sydney, Kensington, Australia, 3 Tropical Diseases Research Centre, Ndola, Zambia, 4 Department of Disease Control, Faculty of Infectious and Tropical Diseases, London School of Hygiene & Tropical Medicine, London, United Kingdom, 5 Bern University of Applied Sciences, School of Health Professions, Midwifery Division, Bern, Switzerland, 6 Institute of Social and Preventative Medicine, University of Bern, Bern, Switzerland

* a.kelly@unsw.edu.au

**Data Availability Statement:** We will not be able to make fully available the data sets underlying our

## Abstract

Curable sexually transmitted infections (STIs) are neglected in public health policy, services and society at large. Effective interventions are available for some STI but seem not to be prioritised at global, regional or local levels. Zambia and Papua New Guinea (PNG) have a high burden of STIs among pregnant women but little is known about the prioritisation of STI treatment and care among this group. We undertook a qualitative study to explore how STIs are prioritised among pregnant women in local health systems in Zambia and PNG. Semi-structured interviews were conducted with 19 key informants—health care workers providing antenatal care, and policy and programme advisers across the two countries. Audio recordings were transcribed and translated into English and stored, managed, and coded in NVivo v12. Analysis used deductive and inductive thematic analysis. Findings were coded against the World Health Organization health system building blocks. Participants spoke about the stigma of STIs at the community level. They described a broad understanding of morbidity associated with undiagnosed and untreated STIs in pregnant women. The importance of testing and treating STIs in pregnancy was well recognised but many spoke of constraints in providing these services due to stock outs of test kits for HIV and syphilis and antibiotics. In both settings, syndromic management remains the mainstay for treating curable STIs. Clinical practice and treatment were not in alignment with current STI guidelines in either country, with participants recognising the need for mentorship and in-service training, as well as the availability of commodities to support their clinical practice. Local disruptions to screening and management of syphilis, HIV and other curable STIs were widely reported in both countries. There is a need to galvanise priority at national and regional levels to ensure ongoing access to supplies needed to undertake STI testing and treatment.

findings. To do so would breach the ethics from each of the participating institutions. However, additional information can be made available on request from the secretary of the Institutional Review Board of the Papua New Guinea Institute of Medical Research, on an individual basis at info@pngimr.org.pg In Zambia this study was approved by the Tropical Disease Research Centre Ethics Committee (TRC/C4/10/2020) and Zambia's National Health Research Authority (NRHA0000012/11/2020). In PNG the study was a sub-study of the WANTAIM trial, which had ethical approval from the PNG Institute of Medical Research Institutional Review Board (IRB#1608), the PNG National Department of health's Medical Research Advisory Committee (MRAC 16.24) and the Human Research Ethics Review Committee at UNSW, Australia (HREC 16708).

**Funding:** This study received funding from the Swiss Network for International Studies, the Swiss Programme for Research on Global Issues for Development (Swiss National Science Foundation grant number IZ07Z0- 160909) (NL) and the Joint Global Health Trials scheme (UK Department for International Development, Medical Research Council, Wellcome Trust; MR/N006089/ (WANTAIM; AV) and MR/S004998/1 (ASPIRE; RMC). The funders had no role in the study design, data collection and analysis, decision to publish, or preparation of the manuscript.

**Competing interests:** The authors have declared that no competing interests exist.

## Introduction

Every day there are more than one million new cases of four curable sexually transmitted infections (STIs)–gonorrhoea, chlamydia, trichomoniasis and syphilis–of which 90% occur in low- and middle-income countries (LMIC) [1], settings that are least able to respond. Untreated STIs are important causes of reproductive morbidity, and are associated with a number of adverse pregnancy and birth outcomes, including stillbirth, premature rupture of membranes, premature birth, low birthweight, neonatal sepsis, pneumonia, neonatal conjunctivitis, and congenital disease [2–6]. Unlike HIV, these infections have not galvanised the political attention and responses that other infections such as HIV has received, as evident, for example with the formation of the Joint United Nations Program on AIDS, the establishment of significant donor programmes including the U.S. President's Emergency Plan for AIDS Relief and the the Global Fund to Fight AIDS, TB and Malaria. Agenda setting is critical to prioritisation [7].

Highlighting their lack of political priority at the global level, as with the Millenium Development Goals (MDGs), the Sustainable Development Goals (SDGs) fail to highlight STIs, and they remain invisible [8]. Despite not being explicitly mentioned, Galati [9] argues that sexual and reproductive health gained greater prominence in the SDGs than it had previously. The challenge, suggest some scholars, lays in the global community identifying important points of convergence between these development goals and STIs, particularly as they relate to health and ending inequalities [10].

The SDGs served as a catalyst for political attention around syphilis and viral hepatitis. And, although elimination of congenital syphilis was the first elimination goal in vertical transmission, the prevention and later elimination of vertical transmission of HIV has received greater attention and focus. In 2016 the World Health Organization (WHO) set ambitious targets for what is now commonly referred to as 'triple elimination'–the elimination of mother-to-child transmission of HIV, syphilis and hepatitis B [1]. Apart from HIV and syphilis, the response to other curable STIs has not attracted global attention, focus and priority including human and fiscal resources; consequently, curable STIs have not been prioritised for prevention, aetiological diagnosis and treatment.

In 2021 WHO declared the global response to STIs was in crisis [11], reporting the uneven progress in global responses to HIV, viral hepatitis and STIs [12]. Now more than ever, STIs are being discussed in global forums, guidelines and strategies, including with combined global health sector strategies on HIV, viral hepatitis and STIs [1, 12]. At a country level there is some indication of prioritisation of STIs, with many countries adopting updated STI treatment guidelines and the monitoring of gonococcal antimicrobial resistance [13]. Despite these indications of prioritisation, other indicators, such as aetiological testing of STIs, is still far from the norm in LMIC settings which have the greatest burden of these curable STIs. Outside of the "triple elimination" agenda, it is unknown how, if at all, curable STIs are being prioritised locally among pregnant women in antenatal care.

Against this backdrop of prioritisation of STIs in maternal and child health, and as part of a larger multidisciplinary study on the political prioritisation of STIs [14], we used the platforms of two clinical trials, one in Zambia [15] and the other in Papua New Guinea (PNG) [16]. These two clinical trials were, in whole or in part, investigating the role of STIs in pregnancy and adverse maternal and newborn health outcomes. Our study aimed to understand ways in which STIs in pregnancy are prioritised, understood and addressed at the local level. Zambia and PNG were chosen because of our ongoing research in these two countries. In both study sites the prevalence of curable STI among pregnant women attending antenatal clinics is high, around 35% in Zambia [17] and 43% in PNG [18]. In contrast, Zambia is considered a high

HIV burden country with HIV prevalence among adults 15–49 years estimated at almost 11% whereas in PNG, HIV prevalence is low, at 1% for the same population [19], but as high as 20% among female sex workers (11–20%) [20].

In this paper we explore how STIs are prioritised in the care and treatment of pregnant women in local health systems. The antenatal clinic offers a unique opportunity to examine critically how STI services are prioritised and provided to reduce the adverse outcomes of curable STIs for pregnant women and their babies. We present and analyse our data using the expanded WHO building blocks for health systems [21].

## Materials and methods

Semi-structured interviews were conducted with 19 health care workers and key informants to explore the treatment and care of pregnant women with STIs, socio-cultural issues and other factors affecting the diagnosis and treatment of STIs in pregnant women, and the impacts of prioritisation of STIs in pregnancy. Semi-structured interviews are qualitative in nature and for research with key informants this method was deemed appropriate as it allowed us to enquire about specific issues, but allowed the opportunity for the informants and interviewers to address other issues as they emerged. Semi structured interview guides were used for all the interviews.

Health care workers included those working in rural and district health facilities in rural and urban settings in the study sites in the two provinces (one in Zambia, one in PNG). Key informants included senior technical advisers working in policy and programme management (Table 1) at the national level. Three interviews were conducted on-line (two clinical health care workers and one programme advisor); 16 were conducted face to face. Interviews were conducted between 8th July 2021 and 31st May 2022, during the height of the COVID-19 pandemic when many primary and public health services were severely affected [22–24].

Interview guides were designed by the lead researchers in Zambia and PNG (KK, AKH). Teams of experienced and highly skilled qualitative researchers from the Tropical Diseases Research Centre, Ndola, Zambia and the PNG Institute of Medical Research were trained together via online platforms (KK, AKH) as well as face-to-face training sessions in each country in preparation for data collection. Interview guides were refined by the team throughout the training and during the initial stages of interviews with ongoing support and discussion between the investigators and the qualitative field team.

Following written informed consent, all interviews were digitally audio-recorded in a private location at the health facility or office. Interviews were conducted by trained researchers; each interview was between 50–75 minutes long. In each country audio recordings were transcribed and, as necessary, translated into English for analysis and then stored, managed and coded in NVivo v12. Translation was conducted by experienced qualitative researchers in the respective countries. All translations were cross-checked for quality and accuracy.

**Table 1. Basic demographics.**

|  | Zambia (n = 8) | Papua New Guinea (n = 11) | Total (n = 19) |
|---|---|---|---|
| **Gender** | | | |
| Woman | 4 | 9 | 13 |
| Man | 4 | 2 | 6 |
| **Informant Type** | | | |
| Health care worker | 2 | 4 | 6 |
| Health care worker / Specialist | 1 | 5 | 6 |
| Policy and programme advisor | 5 | 2 | 7 |

### Sample, recruitment and data collection

Participants were purposively selected from clinical, public health and sexual reproductive and health technical advisory groups to provide detailed emic accounts of STIs in pregnancy. Participants included those working at district and rural health facilities at our study sites. Key informants were selected based on their specialist knowledge relating to maternal health and/ or STI programmes, working at the national level. In both PNG and Zambia interviews were conducted until data saturation was met, whereby no more new information was emerging. Across the two study sites we conducted interviews health care workers, key informants and women attending for antenatal care. In this paper, we report on data derived from semi-structured interviews conducted with health care workers and key informants.

### Data analysis

All interviews were coded and analysed by two experienced researchers. Thematic analysis used a deductive and inductive approach, whereby each transcript was reviewed for data pertaining to the themes prioritised in the interview schedule. Inductive analysis allowed for an exploration of meanings and experiences. A third and final level of coding was developed against the WHO health system building blocks [25, 26]. The original WHO framework had only six blocks, but like De Savigny and Adam [21], we argue for the need of a seventh–people and community. We place them at the centre of the health system (Fig 1) and report our findings through the voices of the health care workers and key informants who took part in the study. Without this seventh block, the building blocks focus disproportionately on the supply side of a health system and not the demand side; people and community play an important role in assessing the effectiveness of a health system [27, 28]. We present our findings for each building block, although they are understood to blur and overlap in the everyday operations of a health system.

### Ethics

In Zambia, the study was approved by the Tropical Disease Research Centre Ethics Committee (TRC/C4/10/2020) and Zambia's National Health Research Authority (NRHA0000012/11/ 2020). In PNG, the study was a sub-study of the WANTAIM trial, which had ethical approval from the PNG Institute of Medical Research Institutional Review Board (IRB number 1608), the PNG National Department of Health's Medical Research Advisory Committee (MRAC 16.24) and the Human Research Ethics Review Committee at UNSW Sydney, Australia (HREC 16708).

## Results

In total, 19 health care workers and key informants were interviewed, eight in Zambia and 11 in PNG (Table 1). Of interviewees, 12 were health care workers providing antenatal and STI care and seven were key informants providing policy and programmatic technical support. More than half of the key informants were women, and most of the men interviewed were from the study in Zambia.

### People and community

In the expanded WHO building blocks framework, people and community are identified as central to the health system [21]. All participants shared biomedical knowledge of STIs, but many reflected that the knowledge in the community drew on different socio-cultural sources of information for interpreting STIs and these differences caused challenges.

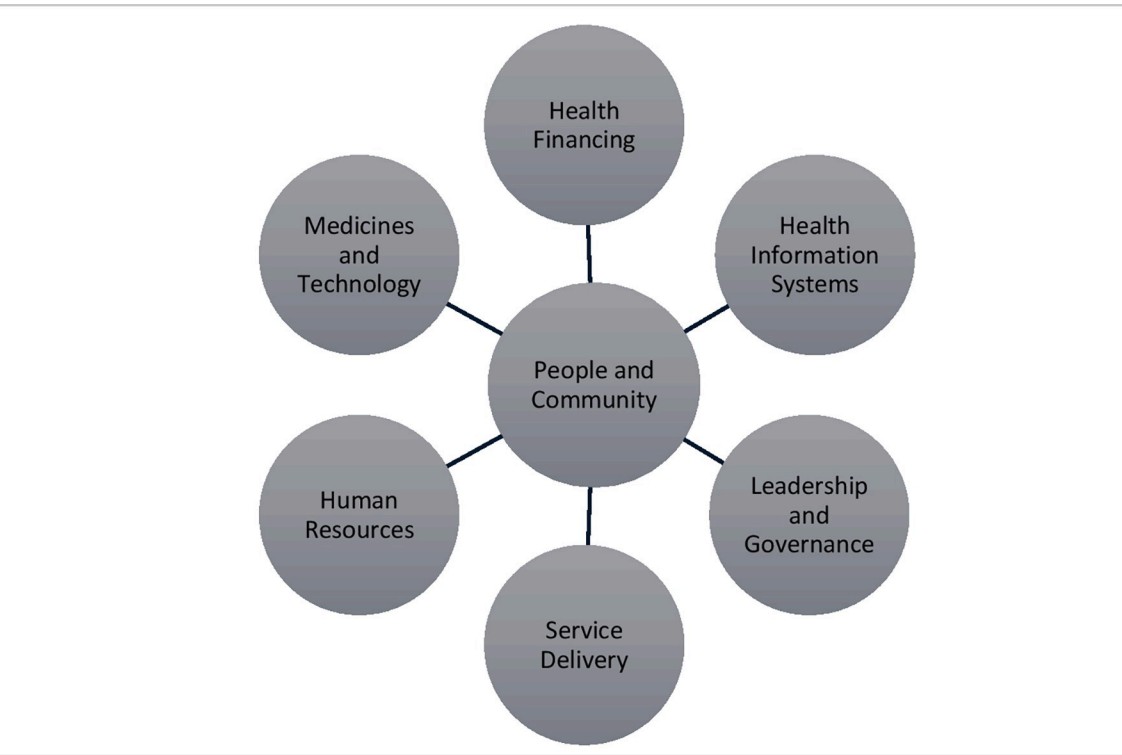

| Service Delivery |
|---|
| Effective and well-functioning health service delivery, comprising: service comprehensiveness, accessibility, coverage, continuity of care, responsive and person-centred care, effective coordination, and management and accountability |
| **Human Resources** |
| Knowledge skills, motivation, and deployment of the workforce responsible for managing and delivery of health service, including qualified health professionals with the appropriate skill set for the context; and the retention, professional development, and clinical support of staff |
| **Medicines and Technology** |
| Equitable access to essential medical products, vaccines and technologies of assured quality, safety, efficacy and cost effectiveness, and their scientifically sound and cost-effective use |
| **Health Financing** |
| Mobilization, accumulation and allocation of finding to cover the health needs of the people, individually and collectively, in the health system |
| **Health Information Systems** |
| Overall data quality, relevance and timeliness; conversion of data into information for health-related decision making |
| **Leadership and Governance** |
| Incorporates policies, accountability, stewardship and partnerships that intersect with all other components of the health system |
| **People and Community** |
| The critical involvement of people and community in health services and programs improves effectiveness of health interventions. People influence all and each of the building blocks |

**Fig 1. The seven building blocks to support the health system [21] and key components of each building block [25, 26].**

Heath care workers discussed a number of issues relating to providing information and education at the community level, including the importance of providing STI awareness and information at the antenatal clinics. Navigating stigma and traditional beliefs about the cause and symptoms associated with STIs was also identified. In some communities, STIs were mentioned as being caused by external forces such as witchcraft. They spoke therefore of needing to challenge traditional beliefs held in the community, especially the reliance on traditional and customary healing practices, causing delays in pregnant women (and others) seeking biomedical treatment.

> *For pregnant mothers with STI, most of them are often found to have problems with vaginal discharge, itchiness, and sore[s]. Most of them believe in sorcery and so do not come quickly. . . they will take herbs, or people work on them back in the village until they could no longer able to [cure], so they come out and we check them. If I check them and find out that they have problem, I talk to them, provide good advice and tell them about the cause, how she got it and that her partner must also be treated for them to be cured from the disease.*

Woman, Sister in Charge STI/HIV, District Hospital, PNG

Study participants also spoke of the need to be open to talk about STIs, in particular because of the effect that untreated STIs could have on the unborn baby. It was recognised that there was a dearth of community awareness on STIs generally and in pregnancy specifically; the opposite was true for HIV. Some of the study participants questioned why information on STIs and STI prevention could not be incorporated into community HIV awareness as the same interventions, such as using condoms, would protect people against both.

Fear of being blamed and accused by a male partner for her infection, as a result of having an extramarital relationship, was a barrier for some women to attend antenatal clinic and be screened for an STI. To avoid stigma and discrimination from their own partners and communities more broadly, some participants described how women, knowing that they have an STI may attend for their antenatal care outside of their catchment area, allowing them to be cared for and managed anonymously. In PNG, one participant spoke of the importance of integrating services to avoid such stigma.

> *. . .it's just among all the services because if you want to prioritise it, like we may cause stigma, so now it's kind of integrated, you know all the services*

Woman, Health Extension Officer and Family Health Services Coordinator, PNG

Apart from the infections themselves, other implications of an STI diagnosed in pregnancy relate to community and individual assumptions about sexual morality and includes gender-based violence. In Zambia, the intersection between STIs and the risk of gender-based violence were identified as an adverse impact of STIs in pregnancy. This risk of violence following a diagnosis of an STI, particularly in pregnancy, reinforces the importance of counselling as part of partner notification. In Zambia, a couple of key informants even spoke of re-testing a woman already known to have syphilis with her husband to avoid the potential blame and violence she may experience should she herself disclose her diagnosis.

> *I think it is a fear of GBV [gender-based violence] issues. . .yeah, because if I tell him, he will think I am immoral. . .Like promiscuous or he will think I am accusing him of being promiscuous I think like that. . .the thing of disclosure of my STI status because of fear of the husband.*

Man, Public Health Specialist Provincial Health Office, Zambia

## Service delivery

Participants identified a number of barriers experienced by women accessing health care facilities for testing and treatment. They included attitude of health workers, cost, travel, and personal costs for purchasing treatment.

Respectful care is essential to quality maternal and newborn care, as outlined by WHO standards of care [29]. The absence of this respectful care, including being stigmatised, being spoken to in harsh ways, and breaches in confidentiality were identified in this study as deterrents for women to engage in STI services during pregnancy. Respectful care was identified as critical to supporting pregnant women to attend clinic and to complete their treatment to prevent adverse outcomes for their babies. One nurse in PNG described the importance of respectful care when she manages pregnant women with an STI.

*When I come and discover that a pregnant mother has this sickness (STI), I would sit her down and explain about her sickness, then give her medicine and explain to her, "You must go and drink all the medicine. . .I am giving you and if you don't it drink on time, sorry for the unfortunate baby. It's your blessing for God who is inside. So if you don't drink medicine then the bacteria will multiply in the child and the medicine won't fight the bacteria. Whatever medicine you take would not work and you will destroy the new temple, the baby inside who came about as the result of you and your husband's love. Therefore, you must drink/take medicine faithfully on time. So I would like you to return back after a week. Or [if] you discuss with your husband, it's good if he drinks the medicine. If he does not drink it, come and let me know and tell him to come and I will discuss with him.*

Woman, Sister in Charge STI/HIV District Hospital, PNG

Women diagnosed with an STI during pregnancy require counselling, education and treatment support to understand the possible impacts of an untreated infection on their unborn baby, the importance of treatment completion, and prevention of re-infection during pregnancy. In the same way that partners have long been considered key in HIV counselling and testing [30], so too are partners in the management of curable STIs in pregnancy. The importance of couple counselling for STI management was identified as paramount in both settings to avoid blame and ensure men were treated alongside their partners. Partner treatment was considered central in both settings to reduce women's risk of becoming re-infected in her pregnancy with an STI following her treatment. Men are considered central to pregnant women's health, and that of their baby.

*We do not only treat the women, but we also treat their partner. . . It does not make sense where only the woman is treated and treated alone, and then go back and have sexual intercourse with a partner that is infected. And if we find couples that are untreated we advise them to refrain from having unprotected sex, or we provide them with condoms so that they can prevent themselves from the re-infection. . . we provide support and care. . . counselling. . . and we also talk about the aspect of transferring the infection to the unborn baby, they need to understand the impact of having an STI while pregnant and the effects of the infection to the unborn baby.*

Woman, Chief Safe Motherhood Advisor, Zambia

## Human resources

The health care workforce and others who resource the health system are critical to effective, responsive and resilient health systems providing maternal and newborn health services. In terms of their level of knowledge and skills in managing and treating pregnant women with STIs, participants spoke in depth about the need and importance of STI management in pregnancy. Syphilis was by far the most common STI discussed and the risk it poses in pregnancy —if untreated—for neonatal as well as maternal health outcomes was discussed. In Zambia, other STIs identified included gonorrhoea and chlamydia; in PNG, syphilis and gonorrhoea, with chlamydia and trichomonas were also noted.

Numerous participants interviewed shared knowledge about clinical practice and STI treatment that highlighted a gap in professional development with much of their information out of date. In PNG for example, a number of participants spoke of wanting to revert back to doing cervical examinations of pregnant women as part of their STI screening. Current PNG national guidelines do not include speculum examination of antenatal women for STIs, as it is now recognised that STIs cannot be identified clinically [31]. Moreover, speculum examination by inexperienced staff may cause problems with the pregnancy. This change in practice, however, still concerned some health care workers who felt that they were failing to identify STIs in pregnant women and treat accordingly, and was therefore the reason for poor pregnancy outcomes.

*I saw that it was very good when they [pregnant mothers] came for speculum examination during pregnancy where many mothers when giving birth gave birth well but if you access the statistics there, you will realise that there are many FDIU cases nowadays, maybe due to STI.*

Woman, Sister in Charge STI/HIV District Hospital, PNG

The absence of ongoing professional training to ensure clinical practice is current shows a lack of prioritisation of bacterial STIs. Others recognised that training was out of date. Staff needed to be updated; frequently the only training they had received was during their formal training. One informant in PNG stated that STI training was conducted by the National Department of Health (NDoH) in 2018 and that there were changes in management of STIs since then, but there had been no dissemination of new guidance that they were aware of. In Zambia, one participant recommended training to improve staff knowledge about the management of STIs during pregnancy, including mentorship by staff who are experienced in the treatment of STIs in pregnancy and having treatment algorithms available in the clinic that relate specifically to pregnancy.

*For example we cannot give tetracycline to treat chlamydia in a pregnant woman. . . One facility we found that they were not conversant on how to manage STIs in pregnant woman, so we need to have that flow chart to follow which has information on syndromic management of STIs and that is when we realised that it was generalised to everyone they have dealt with, they do not specifically look at pregnant woman.*

Woman, Chief Safe Motherhood Advisor, Ministry of Health, Zambia

Some services spoke of designated staff at their facility to provide STI and HIV testing and treatment with the same individual being responsible for providing care to antenatal women. However, other facilities described the difficulties with providing specialised services when there was only one nursing officer responsible for a number of different areas of care, such as malaria, antenatal care and TB care and treatment.

*She's talking multi-tasking. . .talking TB and talking antenatal, and malaria you see. That's the problem in this province with health services.*

Woman, Health Extension Officer and Family Health Services coordinator, PNG

## Essential medicines and technology

Without access to essential medicines and technology there can be no action on a priority health issue. In PNG, testing for HIV and syphilis is included within standard guidelines for all women at their first antenatal visit. However, in reality these opportune times to test are missed due to ongoing supply issues with commodities.

*We used to screen all the antenatal mothers and we treat them and their partners, previously we used to give a single dose of benzathine. But now, we don't have the supply, there is no consistency of syphilis test kit, we are no longer treating all/everyone on according to whatever symptoms they come in but it's just the oral gono-pack [standard treatment pack for gonorrhoea] that we give for all STI patients. . .*

Woman, VCT Sister in Charge and Unit Manager, provincial hospital, PNG

In Zambia, repeat testing for both HIV and syphilis is expected during the pregnancy. Despite this, screening did not always happen, with testing for syphilis routinely suspended due to a shortage of rapid diagnostic test kits. Recognising that shortages in supplies was disrupting the testing and management of syphilis, one senior health officer in Zambia identified that screening for syphilis and HIV was previously a "must" for all newly enrolled women and to not test was referred to as a "cardinal" sin.

*Testing for STIs in pregnancy was a must. You could not get away without testing a woman when they came to register for their antenatal care. . . and the supplies were always there.*

Man, Sexual and Reproductive Health Programme Specialist, Zambia

Even when access to the test kits is possible, stockouts of syphilis treatment means women are diagnosed but go without treatment; the latter of which was described as frequent in Zambia. In these situations the failure to treat women diagnosed with an infection acts as a barrier to their, and other women's, engagement in antenatal care.

*The testing itself and the availability of drugs, they get discouraged. . . because you find that we have the test kits and I test them, "you have STIs", and then I tell them,*

"I don't have drugs. . ." so they are discouraged.

Woman, District Nursing Officer, Zambia

Although prioritised by their inclusion in antenatal care guidelines, ongoing disruption to screening and management of syphilis was widely reported, severely limiting the extent to which pregnant women can access testing (and even treatment).

While STI screening and management may be considered core priorities in antenatal care, the ability to test and treat is reliant on the ongoing supply of diagnostic test kits and essential medicines. Chronic disruptions to the supply of essential STIs medicine and diagnostic test kits were identified in both countries. In some cases the supply of test kits was erratic and

meant while some women may have been tested earlier on her pregnancy, she could not be re-tested, as per the national protocol in Zambia. These disruptions were not confined to the facility or indeed a district. In PNG, the disruption in the supply of syphilis test kits was felt due in part to how supplies and logistics are managed, with the transition to Provincial Health Authorities resulting in changes in management and finance structures.

> . . .when the syphilis kit ran out they said that, 'it's nationwide [and] it's beyond our control.

Woman, STI and HIV counsellor, rural health centre, PNG

Across all participants in each setting that HIV was prioritised over STIs and, as such, there is a better supply of HIV test kits and HIV medication. This led to several participants in both settings making a call for the dual HIV and syphilis test kits to be made available and to circumvent the ongoing supply chain issues with syphilis test kits. The availability of the dual HIV/syphilis test kits was recognised as important to be made available and overcome the ongoing supply issues associated with syphilis test kits.

In Zambia, there appeared to be widespread and ongoing stockout of benzathine benzylpenicillin for at least a year, with treatment for gonorrhoea also disrupted. This became a deterrent for women, and an additional expense, coming back to the clinic to see if the prescribed treatment was available. In cases where the clinic knew they would not have the treatment they instructed women to purchase the medicine themselves, shifting the financial cost of syphilis treatment to pregnant women. Sustainability and availability of drugs to treat STIs is an ongoing concern to encourage access. The adverse outcome of these stockouts was widely discussed by participants, particularly in relation to stillbirth associated with syphilis. A similar issue was discussed in PNG where participants described how only syphilis and HIV are prioritised, and that the ongoing stockouts of syphilis test kits is likely associated with observed increases in perinatal mortality.

> I would advocate for rapid scale up of the dual test kits, the use for dual test kits—testing for HIV as well as STIs, because HIV does not kill the child but syphilis would definitely kill the child.

Woman, Chief Safe Motherhood Advisor, Ministry of Health, Zambia

## Health financing

Whether financed through national programmes or donors, the availability of funding speaks to where health priorities are made, what services are provided and what pregnant women can be tested for. The lack of health financing for STIs was raised as a serious issue affecting the ability to provide STIs services to pregnant women (and their partners) and prevent stillbirths and other poor neonatal health outcomes. Numerous participants in both Zambia and PNG noted the ongoing availability of essential medicines for HIV, TB and malaria, the three infections covered by the Global Fund to Fight AIDS, but not STIs. The drug shortages raised in *Essential Medicines and Technologies* are the adverse outcomes of inadequate domestic health financing and investment from international donors to meet or co-finance these costs.

The inadequate health financing of services, evident through stockout of syphilis tests kits and treatment for STIs, means that the economic burden of STI screening and treatment is passed on to pregnant women and their families, who are often unable to absorb such costs. Examples of how these costs became the burden of pregnant women included being referred to another clinic to undergo screening in case of a shortage of rapid diagnostic tests, being sent

to a private pharmacy to purchase STI treatment in cases where there were no essential STI medications or being asked to return to the clinic to see if essential STIs medicine were available in the future. These costs, and even fear of these costs, becomes a deterrent for pregnant women.

> . . .they may be transferred or referred to some other facilities, so they may incur other transport costs and sometimes the provider may opt to actually give the prescription to the client or the woman to go and drugs from a chemist. So they may incur some cost in-terms of procuring that drug which is not available, but otherwise ministry of health. . . their targets. . . their target is treatment of STIs for pregnant women and when you look at our health centre kit . . .our health centre kits has the test kits for STIs and it also contains the drugs that are supposed to be provided for those women that test positive STIs.

> Woman, Chief Safe Motherhood Advisor, Ministry of Health, Zambia

## Health information systems

There was very limited discussion on health information systems. In the data from PNG, participants did not refer to the need or the absence of epidemiological data on the burden of STIs in pregnancy. In Zambia, however, there was a call to build the evidence base needed to advocate for the prioritisation of STIs, both in the general population and pregnancy. Two participants described the need for syphilis test kits in order to test and treat women to reduce the risk of adverse outcomes, in particular stillbirths which were recognised as increasing in the country and felt to be due to lack of testing for syphilis in pregnancy.

> . . .when a woman delivers a stillbirth we test the them for syphilis. . . it is mandatory they need to be tested for syphilis unless the test kits are not available. . . For those who we test the majority have come out positive for syphilis, so it is contributing to the high

> perinatal mortality that we are seeing.. . .

> Woman, Chief Safe Motherhood Advisor, Ministry of Health, Zambia

What was clear, from both PNG and Zambia, was a broad understanding of morbidity associated with undiagnosed and untreated STIs in pregnant women and associated STIs with a poor outcome for newborns. By far the most commonly discussed adverse outcome of an untreated STI in pregnancy was stillbirth. In Zambia study participants went further, discussing macerated stillbirths or other consequences of syphilis. In both Zambia and PNG participants mentioned other adverse outcomes including infertility, miscarriage and intrauterine growth restriction and preterm birth.

> So, stillbirth seems to be a very big problem but also neonatal mortality, those that are born with STIs and they end up dying. So, babies dying early in their life and babies dying during labour,

> during pregnancy seems to be very big problems.

> Woman, Chief Safe Motherhood Advisor, Ministry of Health, Zambia

In trying to avoid these outcomes, in line with WHO antenatal care recommendations [32] most participants spoke of testing and treatment for HIV and syphilis. In both Zambia and

PNG, HIV and syphilis testing is designed to be conducted using rapid diagnostic test kits at point-of-care to initiate same day treatment and facilitate partner notification, testing and treatment.

In Zambia, testing for viral hepatitis happens at the main hospitals. In PNG, there is no routine testing for viral hepatitis. The draft national guidelines for "Triple Elimination" include rapid diagnostic test for hepatitis B as part of antenatal care, but at the time of the study had not been finalised, as mentioned by one of the senior specialists interviewed.

*. . .for pregnancy we routinely test for syphilis and HIV, those the only two tests that we do. Recently hepatitis B has been introduced for testing but we just we are sort of at the survey stage at the moment, so HIV and syphilis are the ones we test for.*

Woman, Senior Medical Specialist, tertiary hospital, PNG

## Leadership and governance

It was not surprising considering when this data was collected (2021–2022) to hear that COVID-19 was being prioritised as the number one health priority in both Zambia and PNG, with some sharing their frustration that other essential health services were neglected. The pandemic aside, there was a lot of discussion about the extent to which STIs are–or are not–prioritised. It was widely shared that as a viral STI HIV is prioritised over bacterial STIs, and that when referring to STIs, it was syphilis that was a priority service.

In PNG, where the Global Fund to Fight AIDS, TB and Malaria national HIV grant is a joint HIV and TB one, we can see that in this setting at least HIV and TB were now being jointly prioritised, as have HIV and syphilis. Participants in PNG also expressed a lack of commitment and input from the National Department of Health, due to re-structuring of the department, and partly due to staff shortages.

*HIV is prioritised. It has its own basket. And now TB, TB/HIV is put together.*

Family Health Services is a priority for immunisation especially nutrition

Woman, Health Extension Officer and Family Health Services coordinator, PNG

From the policy and health systems perspectives, in terms of reliable consumables and health workforce training, key informants believed that STIs should be given greater priority. A few participants in Zambia at least recognised that at a practical level, in the clinic, STIs were not always the most pressing concern–priority–when compared to other conditions that women may present with, and which left untreated may cause maternal death. An example given was that of diarrhoea in pregnancy. This raised the difference between what may be described as political priority and a clinical priority. But as the participants also explain, the reverse is equally concerning. When STIs in pregnancy are a clinical priority–and testing for STIs in recommended in both countries–it cannot be prioritised because the consumables needed to undertake the testing are not available, nor are the drugs needed to provide treatment and cure the infection, as is the case with gonorrhoea and syphilis.

In Zambia the inclusion of STIs in the guidelines on pregnancy was viewed as evidence of prioritisation, with several participants noting that hepatitis is now included in the 2020 national guidelines for its adverse effects on women during pregnancy including severe liver disease and death.

Across all interviews, participants described that while STIs are an important feature of antenatal clinics, with visible and profound impacts, particularly on neonatal outcomes, is not

prioritised in the way that other diseases affecting pregnant women are. The most common comparator to the prioritisation of STIs was HIV, a virus that is readily identified also as a STI. To this end, some suggested that HIV should be part of the STI care and treatment provided in antenatal clinics, not as separate services.

*It should be the other way around, HIV is a sub-programme within the STI programme*

Man, Public Health Specialist Provincial Health Office, Zambia

## Discussion

Despite widespread agreement that there is an association between bacterial STIs and adverse maternal and newborn health outcomes [2–6], it has been widely purported that these infections have not been prioritised for prevention, aetiological diagnosis and treatment. Declared neglected, and in crisis [11], the global response to STIs has been brought to the fore. It is in this context that we sought to understand and examine the prioritisation of STIs in pregnant women in antenatal clinics in Zambia and PNG, two high burden settings for STIs. Using the expanded WHO building blocks conceptual framework as described by De Savigny and Adam [21], we have detailed the various areas of the health system that address and respond to STIs and the efforts of health care workers and others to address STIs in their services within two different socio-cultural settings. We found overall, that health care workers and health policy and programme advisors viewed curable STIs as being neglected. While COVID-19 has gravely impacted primary health services since 2020, COVID-19 was not identified as the cause of such neglect described in these two countries.

Participants in both countries spoke in detail across most aspects of the health system, detailing the interconnectedness of these elements and how these impact on the prioritisation of STIs in clinical care. All participants were knowledgeable about morbidities associated with undiagnosed and untreated STIs in pregnant women such as preterm birth, growth restriction, neonatal conjunctivitis and stillbirths, all of which were identified by those interviewed for the study. In order to prevent these adverse outcomes all participants spoke of the clinical imperative to test for curable STIs in pregnancy, provide treatment, provide partners testing and treatment and avoid reinfection during pregnancy. Whilst clinical guidelines are available to provide up to date testing, treatment and care information, some health care workers raised issues about the disjuncture between policy level changes (i.e. new guidelines) and the failure to socialise the guidelines through training; others felt that the guidelines for STI management were not specific enough for pregnant women. Further still, others objected to current best practice and wanted guidelines to be reversed to return to pelvic examination to identify an STI.

Even where the policies and frameworks clearly prioritise syphilis, the only curable STI to be prioritised in this way, pregnant women and their babies were not able to benefit from this level of prioritisation. Without the funding to procure the rapid diagnostic test kits, and at least in Zambia ensure the ongoing availability of antibiotics, health care workers were not able to action these guidelines, transforming political prioritisation into quality clinical care in the antenatal clinic. Test and treat for HIV and syphilis is included in WHO guidelines and recommendations for antenatal care [32]. Although prioritised by their inclusion in antenatal care guidelines, ongoing disruption to screening and management of syphilis was widely reported, limiting the extent to which pregnant women can access testing (and even treatment). Ongoing stockouts of rapid diagnostic test kits for syphilis have been noted as a failure

of health systems in a variety of settings [33] and is not unique to Zambia or PNG. But it does raise important questions about how health systems operate and the disjuncture between priorities in guidelines and policy and action. In countries reliant on donor funding for HIV, including HIV rapid antigen test kits, syphilis test kits are usually provided by national governments resulting in multiple procurement and funding mechanisms for test kits designed to be used on the same women in the same service. The shift to the dual HIV/syphilis test kit for use in antenatal clinics, especially for countries where donor agencies oversee procurement and financing will ensure greater and more reliable access to syphilis testing, at least on par with HIV testing. The WHO acknowledges a variety of benefits of the dual test in antenatal clinics, including the added benefit that health care workers will not be required to be trained on multiple testing algorithms [34]. For other bacterial STIs (gonorrhoea, trichomonas and chlamydia) syndromic management remains the main stay for identifying them in both settings.

In neither setting was there any discussion about interest or investment in innovations to introduce STI testing for gonorrhoea, trichomonas and chlamydia and provide targeted treatment for pregnant women. Yet across the board there was a keen interest by the participants in both countries, from health care providers and policy and programme advisors, that STIs in pregnancy should be prioritised. Some suggested that rather than discussing HIV and STIs, the focus should be on STIs more broadly, of which HIV is one.

Much of the discussion from participants focused on the original six WHO building blocks, yet the issues of people and community remained central to the issues at the core of being able to take action on STIs when and where its prioritised. Civil society and grassroots mobilisation are important factors that help to raise the level of priority of health issues [35]. Stigmatisation at the community level likely contributed to the lack of priority afforded to curable STIs locally and globally. Issues of shame and embarrassment for attending STI services continues to impede pregnant women from accessing clinical care, as does the cultural aetiology that many have about the cause of symptoms associated with untreated STIs.

In this study, a lack of commitment at the national level, together with staff shortages at the facility levels was a concern for those providing services on the ground and poor financing of services was evident through stockouts and lack of essential antibiotics to treat curable STIs. In Zambia the inclusion of STIs in the guidelines on pregnancy was viewed as evidence of prioritisation, with several informants noting that hepatitis B is now included in the 2020 national guidelines for its adverse effects on women during pregnancy including severe liver disease and death. The disjuncture in both PNG and Zambia between policies that stipulate STIs screening (in PNG this is for HIV and syphilis) should be done for all women; and the ongoing impediment to enabling the effective implementation of that policy, through the recurrent stockouts of drugs and tests strips and inadequate work force, shows that neither country is currently in a position to address the burden of STIs in pregnant women. It is evident that the financial and logistical priority to ensure ongoing access to supplies needed to undertake testing and treatment has not been harnessed. Finally, it has been shown that addressing the neglect in the treatment of STIs requires a comprehensive and multi-faceted approach. This includes promoting sexual health education and awareness, reducing stigma through public campaigns, increasing access to affordable and confidential testing and treatment, integrating STIs services into primary health care, and civil society advocacy for adequate funding for STI prevention and control programmes.

But how do these concerns translate to being prioritised in an already overstretched and under-resourced health systems? For those in this study, for whom maternal and newborn health is their focus, these infections are also their professional priorities, but are, as yet, not those of global commitments such as the SDGs, country leaders and donors. While prioritised in these commitments and antenatal care guidelines, ongoing disruption to screening and

management of syphilis due to stockouts, and the absence of point-of-care hepatitis B screening in these study settings, illustrates the disconnect between national, regional and global policy priorities and these priorities being programme actionable. As long as these infections remain undiagnosed, and therefore likely untreated, and remain associated with adverse maternal and newborn health, greater efforts are needed to support and galvanise the political priority needed to not only prioritise some infections over others, but ensure that the health system as a whole (including people and communities) remains resourced, resilient and capable to move from policy to action, without which such priorities are mute.

The prioritisation of testing and treatment of HIV and syphilis in pregnancy to improve birth outcomes [34] is well established, however the evidence for antenatal screening of other curable STIs such as gonorrhoea, chlamydia and trichomonas is evolving. A number of systematic reviews and meta-analysis have looked at the impacts of trichomonas [36], chlamydia [37] and gonorrhoea [38] and their association with adverse pregnancy and birth outcomes—premature rupture of membranes [36–38], preterm birth [36–38], low birth weight [37, 38] and stillbirth [37, 38]. These findings on associations with adverse maternal and newborn health outcomes have been moving the global community towards prioritising strategies to provide STI testing and treatment within routine antenatal care to improve newborn outcomes. Recently available data from the WANTAIM trial [16], in which this qualitative work was embedded within in PNG, has provided world first data on the impact of treatment of STIs on mitigating these adverse health outcomes. Among antenatal women, providing point of care testing and treatment for gonorrhoea, chlamydia and trichomonas had no statistically significant impact on preterm birth or low birth weight compared with standard antenatal care; in a sub-group analysis there was reduction in low birth weight among women treated for gonorrhoea. This finding poses a new and different question on prioritisation. The focus on antenatal women and STIs has largely centred on newborn health outcomes, not maternal health, as it was historically with HIV treatment in pregnancy before the adoption of Option B + in the early 2010s. In light if this new data, now we need to ask should we be testing just to improve newborn outcomes, or does this new data provide an opportunity to think into the future about why and how STIs may be prioritised among pregnant women to promote the overall health and well-being of reproductive health more broadly? Participants in our study stated that antenatal care is an important opportunity to provide services and care, among women who may otherwise not attend for health care or know their STI status.

The overall number of those who participated in this qualitative study was modest, but not uncommon in qualitative research, yet we quickly reached data saturation both within each country and across the two and we were no longer generating new information or ideas from the participants through coding. This speaks to the quality of our data and the representativeness of our participants and that a larger sample would not have yielded more data. During the analysis process it was possible to discuss and review the generated codes and categories within and across the countries, increasing the credibility of the results, important indicators of the quality of research findings [39].

## Conclusion

Curable STIs continue to pose a significant burden among pregnant women which, left untreated, have been associated with a number of adverse maternal and newborn health outcomes. For this reason alone these infections are an important public health concern in Zambia and PNG, as articulated by the participants in this study. While there is increasing recognition globally, and in each of the countries included in this study, of the priority of testing and treating pregnant women for the intent of improving her own health and eliminating

the transmission of HIV, syphilis and hepatis B to newborn babies, the same prioritisation has not as yet been afforded to other curable STIs including chlamydia, trichomonas and gonorrhoea. The prioritisation of curable STIs is yet to be actualised among pregnant women, particularly those in LMIC such as Zambia and PNG, despite the recognition of improving health and ending inequalities.

## Acknowledgments

We thank the women who participated in the ASPIRE and WANTAIM Trial and the WANTAIM Study Group of trial investigators and project staff.

## Author Contributions

**Conceptualization:** Eva Cignacco, Nicola Low, Angela Kelly-Hanku.

**Data curation:** Kelvin Kapungu, Alice Mengi, Mike Chaponda, Nicola Low, Angela Kelly-Hanku.

**Formal analysis:** Lisa M. Vallely, Kelvin Kapungu, Angela Kelly-Hanku.

**Funding acquisition:** Nicola Low.

**Investigation:** Nicola Low.

**Methodology:** Eva Cignacco, Angela Kelly-Hanku.

**Supervision:** Angela Kelly-Hanku.

**Writing – original draft:** Lisa M. Vallely.

**Writing – review & editing:** Kelvin Kapungu, Alice Mengi, Mike Chaponda, R. Matthew Chico, Michaela A. Riddell, Andrew J. Vallely, William Pomat, Eva Cignacco, Nicola Low, Angela Kelly-Hanku.

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
