## [Decision Letter · Decision Letter 0]

19 Feb 2024

PGPH-D-23-01976

The prioritisation of curable sexually transmitted infections among pregnant women in Africa and the Pacific: Qualitative insights from health care workers, policy makers and programme advisors in Zambia and Papua New Guinea

Dear Prof Angela Kelly-Hanku,

Thank you for submitting your manuscript to PLOS Global Public Health. After careful consideration, we feel that it has merit but does not fully meet PLOS Global Public Health’s publication criteria as it currently stands. Therefore, we invite you to submit a revised version of the manuscript that addresses the points raised during the review process.

Please ensure that your decision is justified on PLOS Global Public Health’s publication criteria and not, for example, on novelty or perceived impact.

We look forward to receiving your revised manuscript.

Kind regards,

Md Nazmul Huda, BSS, MSS, PhD

Academic Editor

Journal Requirements:

Additional Editor Comments (if provided):

Reviewers' comments:

Reviewer's Responses to Questions

**Comments to the Author**

1. Does this manuscript meet PLOS Global Public Health’s publication criteria? Is the manuscript technically sound, and do the data support the conclusions? The manuscript must describe methodologically and ethically rigorous research with conclusions that are appropriately drawn based on the data presented.

Reviewer #1: Yes

Reviewer #2: Partly

2. Has the statistical analysis been performed appropriately and rigorously?

Reviewer #1: Yes

Reviewer #2: N/A

3. Have the authors made all data underlying the findings in their manuscript fully available (please refer to the Data Availability Statement at the start of the manuscript PDF file)?

Reviewer #1: Yes

Reviewer #2: No

4. Is the manuscript presented in an intelligible fashion and written in standard English?

Reviewer #1: Yes

Reviewer #2: No

5. Review Comments to the Author

Reviewer #1: 1) I think it would be helpful if the paper highlights how the gender of the informants (twice females as males) could possibly (if any) have influenced their responses?

2) Are there any reason other than the prevalence of STIs in selecting Zambia and Papua New Guinea? What are the similarities and differences between these two populations?

Reviewer #2: This paper is informative and explores an important issue but requires major revisions. Below are my suggestions and comments for the authors:

TITLE

The title of the manuscript is very lengthy and confusing. So, it needs to change the title to make short and easy understandable. The title may be renamed as “The prioritisation of curable sexually transmitted infections among pregnant women in Zambia and Papua New Guinea: Qualitative insights”

INTRODUCTION

This section well apprises on curable STIs and the little attention paid to these infections (except HIV). However, this section lacks information on the study’s research question, i.e. how the STIs are prioritized among expecting women who are under antenatal care. Information on this issue are available at global level; the authors are recommended to provide more information so as to better portray the need to explore the research question.

It is written that the two countries, namely Zambia and Papua New Guinea were selected due to different socio-cultural and epidemiological settings. However, there is no mention of how this difference contributes to this research. Recommended to elaborate in this regard.

METHODS

Lines 106-108: Interviews were conducted…………….were severely affected. Interviews were conducted with whom? How many respondents were? Please write one or two sentence about it and refer to Table 1.

Please include the information on study respondents in this section. It is unclear whether the respondents are from rural or urban areas and the types of facilities that the health care workers were employed at. There are examples of health service delivery differing according to geographical areas and different types of health facilities.

Semi-structured interviews. Please make clear about it. Face to face interview? Or in-depth interview…………….?

The weakest area of the paper is the methods section. Without elaborating about following aspects, it is difficult to evaluate the paper:

-Why the two countries- Zambia and PNG were selected? Please provide detail information.

-Purposively selected only 19 participants . What were the selection processes? Nothing has mentioned about this in the paper.

- The respondents were selected from which levels? i.e. national, regional, district, sub-district levels.

-How the sample size was determined from different levels. Please provide detail information.

- Interview guides were co-designed by the lead researchers in Zambia and PNG. What does it mean? Please provide information related to the guidelines used for data collection, i.e. details about the guidelines, type of guidelines developed and used for data collection with different type of respondents? Were the guidelines pre-tested before collecting data?

- Semi-structured interviews were conducted with healthcare workers? But the study title shows that it is a qualitative study. How those matched? Why other methods of qualitative data collection were not applied for data collection which might allow the authors to do data triangulation?

- Please provide information about the data collectors, training, data quality control .

- How data quality were controlled at different stages?

- Key informants were interviews, who were the key informants? Were healthcare workers also key informants?

The sample size is not ample enough to cover the situation of two countries. Population in both Zambia and Papua New Guinea stands at millions whereas only a total of 19 respondents were interviewed from both the countries. It is recommended that the authors shrink down the study areas to regional or local levels according to where the respondents are employed at.

RESULTS

The result section focuses more on challenges, under each theme instead of providing a concrete overview on STIs prioritization among expecting women under antenatal care. It is recommended that they provide a balanced write up. It is also recommended to include some information related to existing policy and knowledge of respondents on existing policies in this regard.

It is good that the results are presented following WHO building blocks, but the presentation could be made more logically. The authors are advised to rewrite the section focusing the key variables comparing between the two countries. A table can be provided with the key common findings in the two countries for easy understanding of the readers.

DISCUSSION

This section appears to be a summarized version of the result section. The authors are requested to re-write this section and focus more on the interpretations and implications derived from their study findings. This will help the authors to portray the significance of their results in successfully addressing their research question.

OTHER REMARKS

The writing is comprehensible but requires improvements. It is recommended that the manuscript be revised for language and grammar.

6. PLOS authors have the option to publish the peer review history of their article (what does this mean?). If published, this will include your full peer review and any attached files.

**Do you want your identity to be public for this peer review?** For information about this choice, including consent withdrawal, please see our Privacy Policy.

Reviewer #1: No

Reviewer #2: No

---

## [Editor Report · Decision Letter 1]

13 Jun 2024

The prioritisation of curable sexually transmitted infections among pregnant women in Zambia and Papua New Guinea: Qualitative insights

PGPH-D-23-01976R1

Dear Dr Angela Kelly-Hanku,

We are pleased to inform you that your manuscript 'The prioritisation of curable sexually transmitted infections among pregnant women in Zambia and Papua New Guinea: Qualitative insights' has been provisionally accepted for publication in PLOS Global Public Health.

Best regards,

Md Nazmul Huda, PhD, MRes, BSS

Academic Editor